# Liver Disease: Induction, Progression, Immunological Mechanisms, and Therapeutic Interventions

**DOI:** 10.3390/ijms22136777

**Published:** 2021-06-24

**Authors:** Sarah Y. Neshat, Victor M. Quiroz, Yuanjia Wang, Sebastian Tamayo, Joshua C. Doloff

**Affiliations:** 1Department of Biomedical Engineering, Translational Tissue Engineering Center, Wilmer Eye Institute, Johns Hopkins University School of Medicine, Baltimore, MD 21287, USA; sarahneshat@jhmi.edu (S.Y.N.); vquiroz2@jhmi.edu (V.M.Q.); wgyuanjia@gmail.com (Y.W.); stamayo2@jhu.edu (S.T.); 2Department of Materials Science and Engineering, Institute for NanoBioTechnology, Johns Hopkins University, Baltimore, MD 21218, USA; 3Sidney Kimmel Comprehensive Cancer Center, Oncology-Cancer Immunology Sidney Kimmel Comprehensive Cancer Center and the Bloomberg-Kimmel Institute for Cancer Immunotherapy, Johns Hopkins University School of Medicine, Baltimore, MD 21231, USA

**Keywords:** mesenchymal stem cells, cirrhosis, fibrosis, alcoholic liver disease, non-alcoholic fatty liver disease, in vivo, clinical trials, immunomodulation, liver, autoimmune disease, hepatitis, cytokines, apoptosis, hepatocyte

## Abstract

The liver is an organ with impressive regenerative potential and has been shown to heal sizable portions after their removal. However, certain diseases can overstimulate its potential to self-heal and cause excessive cellular matrix and collagen buildup. Decompensation of liver fibrosis leads to cirrhosis, a buildup of fibrotic ECM that impedes the liver’s ability to efficiently exchange fluid. This review summarizes the complex immunological activities in different liver diseases, and how failure to maintain liver homeostasis leads to progressive fibrotic tissue development. We also discuss a variety of pathologies that lead to liver cirrhosis, such as alcoholic liver disease and chronic hepatitis B virus (HBV). Mesenchymal stem cells are widely studied for their potential in tissue replacement and engineering. Herein, we discuss the potential of MSCs to regulate immune response and alter the disease state. Substantial efforts have been performed in preclinical animal testing, showing promising results following inhibition of host immunity. Finally, we outline the current state of clinical trials with mesenchymal stem cells and other cellular and non-cellular therapies as they relate to the detection and treatment of liver cirrhosis.

## 1. Liver Fibrosis Overview, Causes, and Burden

The buildup of scar tissue in the liver, or cirrhosis, is a public health concern that affects over 600,000 American adults [1]. The progression of liver cirrhosis is increasingly detrimental as scar tissue accumulates, given it directly interferes with liver function and contributes to gradual liver failure, which can ultimately lead to the death of the individual. Additionally, cirrhosis is difficult to reverse, and doing so would involve either complete liver replacement or regenerating scarred tissue into healthy tissue [2]. Clinically, fibrosis is diagnosed through blood tests that evaluate how well the liver is functioning. These tests look for specific enzymes that might indicate liver failure, such as alanine and aspartate aminotransferase tests (ALT and AST, respectively), which convert proteins into liver energy for liver cells and metabolize amino acids, respectively [3]. These tests also look for biologically important molecules such as creatinine, which is a waste product made by your muscles, or bilirubin, which is the yellow pigment remaining after old blood cells are broken down. A doctor may also order imaging tests that allow them to visualize any abnormalities in liver size or shape [4]. Various diseases cause cirrhosis, but the most prominent cause is liver disease as a result of repeated alcohol overuse. Other diseases that have the potential to induce cirrhosis are non-alcoholic fatty liver disease, chronic hepatitis B, and chronic hepatitis C. Recently, a negative association has been identified between COVID-19 infection and patients with cirrhosis, with a 45% liver decomposition rate during intercurrent COVID-19 [5]. Therefore, there is a critical need for therapies that aim to reverse fibrosis from molecular targeting to harnessing cell engineering techniques.

### 1.1. Alcoholic Liver Disease

Alcoholic Liver Disease (ALD) is caused by excessive drinking and intake of alcohol. Following ethanol ingestion, the organic chemical compound is oxidized to acetaldehyde by a group of enzymes called alcohol dehydrogenases (ADHs). The liver plays a key role in the metabolic breakdown of ethanol to acetaldehyde and subsequent oxidation of acetaldehyde to acetate, and this pathway is kinetically slow in comparison to other biological processes [6]. Acetaldehyde is a known carcinogen, and its presence is detrimental to liver health, often being the cause for ALD. Adducts, known as chemical modifications that could potentially interfere with normal biological processes, can form as a result of ALD. Oxidation of acetaldehyde to acetate also produces reactive oxygen species (ROS) as a byproduct and excessive buildup of these molecules causes oxidative stress, which has been proposed to be crucially involved in ALD. Immunologically, ALD is marked by inflammation and involves both recruited and resident inflammatory cells [7]. Kupffer cells (KCs), which are resident inflammatory cells in the liver, have been shown to be key players in the initiation of ALD when they are inappropriately activated [8]. ALD has also been linked to an increase in cell death through apoptosis. Such apoptotic cell death is often attributed to the increased oxidative stress mentioned earlier and proceeds by going through an execution pathway involving caspases. 

### 1.2. Non-Alcoholic Fatty Liver Disease 

Non-alcoholic Fatty Liver Disease (NAFLD) is the most common chronic liver disease in the world and is present in 30% of the general adult population [9]. An individual’s daily diet and activity level have a significant impact on their likelihood of developing NAFLD, since these two factors are generally a good indicator of body weight. This disease develops in a two-step process. The first step is the development of hepatic steatosis via triglyceride accumulation in hepatocytes. The second step is oxidative stress and proinflammatory cytokine activation that ultimately leads to fibrosis [10]. Some mechanisms propose high baseline levels of adipose tissue which then become inflamed, although the specifics of these mechanisms remain unknown [11]. High baseline levels of adipose tissue leads to insulin resistance, which then gives way to increased adipocyte lipolysis, increased gluconeogenesis, and reduced hepatic glycogen storage [12]. Hyperinsulinemia develops in tandem with insulin resistance and augments hepatic lipogenesis pathways. Hepatic steatosis and triglyceride secretion result from these factors and the increased lipid load spreads to adipose tissue, putting further stress on adipocytes to store these lipids. 

### 1.3. Chronic Hepatitis C 

Chronic hepatitis C virus (HCV) is a blood-borne infection that is transmitted through blood, commonly caused by the exchange of needles or accidental puncture by infected glass or other sharps [13]. HCV can cause chronic hepatitis, which may progress to cirrhosis if not treated correctly. HCV also causes insulin resistance and increases oxidative stress exacerbate steatosis, also presented in NAFLD and ALD. The exact mechanism for this is not fully understood, but it is known that HCV inhibits protein kinase R (PKR), which mediates interferon activity. Interferon activity is historically associated with anti-viral host response, so it would be logical for this to be a key step in the mechanism [14]. Pan-genotypic antiviral agents and direct-acting antivirals (DAAs) that target viral replication are the current standard of care in the clinic but may cause associated risks of developing hepatocellular carcinoma (HCC) [15,16,17,18]. 

### 1.4. Chronic Hepatitis B 

Chronic hepatitis B virus (HBV) is a non-cytopathic virus, meaning that liver damage is thought to be immune-mediated rather than due to direct effects of the virus [19]. To this point, immune antibody responses are heavily dependent on T cells attributed to the virus’ lack of interferon response induction [20]. Natural killer (NK) cell levels increase 10-fold in the early stages of HBV, meaning they are potentially important to initial viral containment [21]. HBV-infected cells are then exposed to constant signals mediated by immune cytokines, growth hormones, lymphocytes, and KCs [22]. This leads to chronic inflammation of the liver, which in turn results in fibrosis. 

## 2. Inflammation and Progression of Liver Fibrosis

The liver has a complex immune regulatory environment that is constantly exposed to products of digestion, environmental antigens, and potential pathogenic molecules or endotoxins secreted by gut microbiota via the portal vein [23,24]. Although merging with the hepatic artery, the portal vein supplies 80% of nutrient-rich blood to liver tissue from the gastrointestinal tract (GI tract) [23,25]. The liver, serving as a front-line immune barrier, must precisely detect, capture and kill foreign pathogens while filtering the rest [26]. This phenomenon is called liver immune homeostasis. A healthy liver has to maintain tolerance to food antigens while protecting itself from pathogens with a controlled inflammation response [27]. Otherwise, if its equilibrium is dysregulated, excessive inflammation would elicit diseases that consequentially lead to liver fibrosis.

Liver fibrosis, and its end stage, cirrhosis, are the common consequences shared across all major chronic liver diseases with activation of HpSCs as the dominant mechanism of fibrotic tissue deposition [28] (Figure 1). Fibrosis is a wound healing defense mechanism initiated by inflammation or injury, but the immune system presence in the liver and the inherent inflammation causing disrupted organ architecture may lead to immunodeficiencies and immune paralysis [29]. Hepatic fibrosis is caused by the excessive production and accumulation of insoluble collagen and extracellular matrix (ECM) components following sustained chronic injury in the liver [23]. Hepatic Stellate Cells (HpSCs) have been identified as effectors of cirrhosis. When activated, HpSCs are associated with fibrotic matrix deposition and fibrillar collagen production. Liver ECM composition is changed through fibrosis, with its structure shifting from laminin and type IV collagen to interstitial collagen (types I and III). This shift continues until the architecture of the liver is significantly altered due to the change in connective tissue composition and neovascularization [30]. The ECM is degraded by a variety of enzymes, but the most prominent type of ECM-degrading enzymes are matrix metalloproteinases (MMPs). MMPs are downregulated by a variety of pro-fibrotic cytokines and are also prone to inhibition by extracellular factors [31]. Although reversing fibrosis is difficult since it involves recomposing the ECM, some studies have had success in reversing it by increasing ECM degradation and decreasing ECM production through an increase in collagenase activity. These studies used exosomes to deactivate HpSCs, remodel ECM production, and inhibit macrophage activation [32]. Understanding the functions and unique properties of the liver is necessary to thoroughly comprehend the mechanisms behind the various diseases that cause cirrhosis.

Constant firing of pro-inflammatory factors in chronic disease from the hepatic immune microenvironment activates HpSCs and myofibroblasts (MFs). Quiescent HpSCs resides in the space of Disse, between neighboring parenchymal cells, hepatocytes and sinusoidal endothelial cells. This cell network is involved with the intercellular transport of soluble cytokines, vitamin A (retinoid) droplet stores, and synthesis of several cytoskeletal proteins. Once activated, HpSCs switch to a proliferative and fibrogenic phenotype with upregulated collagen synthesis activities. In homeostasis, controlled HpSC activation involves secretion of hepatocyte growth factor (HGF), vascular endothelial growth factor (VEGF), tissue inhibitors of metalloproteinases (TIMPs), and the highly profibrotic cytokine transforming growth factor-beta (TGF-β) while at the same time maintaining adequate amounts of proteolytic matrix metalloproteinases (MMPs) to avoid TIMP overexpression [33,34]. However, TGF-β signaling, inflammation and overexpression of myofibroblast-like HpSCs disturb the balance between TIMP/MMP, leading to decreased ECM turnover and collagen type I and II accumulation [35,36].

A variety of pathways result in HpSC activation. In this review, we will mostly be focusing on NK and NKT cell-initiated HpSC activation in HBV patients, cytokines and KC-related non-alcoholic fatty liver disease (NAFLD), as well as T cell-mediated autoimmune hepatitis.

### 2.1. Kupffer Cells and Liver Inflammation in Non-Alcoholic Fatty Liver Disease

Before discussing disease mechanism, a brief introduction of liver cell types is important for later discussions in this review. Hepatocytes, the main functional cells in the liver, perform activities including but not limited to metabolism, nutrient storage, and detoxification [37]. Detoxification involves removing the immunogenic particles through pattern recognition receptors (PRRs), which recognize microbe-associated molecular patterns (MAMPs) and damage-associated molecular patterns (DAMPs). DAMPs, for example, are endogenous proteins released by stressed or injured cells and tissue, that are abundant when the liver is damaged from molecules in the portal vein [23,38]. Kupffer Cells are also involved in PRR signaling and are liver resident macrophages that take up to 90% of the total pre-fixed, resting macrophage population in the body, and up to 15% of the total liver cell population [23,24]. Similar to circulatory macrophages, KCs have the ability to differentiate towards M1 (pro-inflammatory) or M2 (anti-inflammatory) types based on external signals. Interferon-γ (IFNγ), lipopolysaccharide (LPS), and inflammatory cytokines such as tumor necrosis factor-alpha (TNFα) can drive M1 polarization, lead to inflammatory interleukin (IL-1β, IL-12, IL-23) upregulation, and increase the production of nitric oxide (NO) and reactive oxygen species (ROS) [39].

In NAFLD patients, disproportionally high levels of cholesterol, fat, and carbohydrates in the portal vein supply are constantly bombarding first line innate immune populations, promoting constant TLR signaling and pro-inflammatory cytokine secretion [40]. Professional antigen-presenting cells (APCs), such as dendritic cells (DCs) and KCs, would switch from oxidative phosphorylation to aerobic glycolysis, known as the Warburg effect [23], resulting in increased proliferation and cytokine secretion via activation of nuclear factor-kappa B (NF-κB), mitogen-activated protein kinase (MAPK), and extracellular signal-regulated kinase 1 (ERK1). Substantial production of pro-inflammatory cytokines such as IL-1β, IL-2, and IL-8/CXCL8, are paracrine stimuli to HpSC activation [36].

### 2.2. T Cell-Mediated Inflammation in Autoimmune Hepatitis

The liver is also known as a “graveyard” for T cells [23]. Several studies propose that the liver has the function of accumulation and apoptosis of activated CD8^+^ T cells. Once a CD8^+^ T cell is activated it will go through clonal expansion and become distributed to non-lymphoid organs via the blood. During this process, adhesive molecules, such as integrin ICAM-1 will be up-regulated [41]. Once the T cell has reached its desired location, it will contact the endothelial wall, continue to move/roll along the endothelium, and start to attach to the vessel wall. Scoazec et al. found that the liver has both high expression of FasL, a pro-apoptotic factor, and ICAM-1 [42]. Mehal et al. also confirmed that when transfusing resting and activated T cells into wildtype or ICAM-1-deficient C57BL/6J mice, activated CD8^+^ retention was largely decreased in ICAM-1-deficient mice. Retained CD8^+^ T cells remained in close contact with Kupffer cells and began to apoptosis within 14 h [41].

Autoimmune hepatitis (AIH) is a liver inflammatory disease, similar to viral hepatitis, but instead where autoantibodies and debris trigger the immune response. Characteristics such as immune cell infiltration in portal and periportal sites, hypergammaglobulinemia, and autoantibodies are main indications [43]. Alteration of immune tolerance is generally related to changes in cytotoxic T cell activities. While the exact mechanism of AIH is not fully understood, gene mutations induced by environment, drugs, infection, or other factors are potential root causes. For example, the linkage disequilibrium of human leukocyte antigen (HLA) is one of the most observed reasons for Type 1 AIH. HLA is a group of surface antigens that form the major histocompatibility complex (MHC), and which plays an important role in how the immune system distinguishes “self” versus “non-self” antigens [44]. HLA loci were found to be some of the most important genes for infectious disease susceptibility as well. HLA functional polypeptides are coded by thousands of alleles, and their polymorphisms enable the ability to match against the diversity of microorganisms and foreign antigens, but also increases the chance of mutations and of autoimmune disease. Alteration of HLA-B8, HLA-DR3, HLA-DR4, and HLA-DQ2 would lead to activation of Th0-type T cells, B cells, and natural killer (NK) cells, shifting Th0-type T cells predominantly into Th1-type (CD4^+^ effector) and Th17-type (by IL-6 and TGF-β) T cells. Meanwhile, the elevated expression of chemokines, such as chemokine (C-X-C motif) ligands CXCL9 and CXCL10, leads to the attraction of a larger number of immune cells from the local liver and circulation. Chemokine signaling pathways are heavily involved in HpSC modulation as well. For example, CCR5 will recruit monocytes/macrophages and induce HpSC differentiation [45].

### 2.3. Natural Killer and Natural Killer T Cell-Related Inflammation in Viral Hepatitis

Natural killer cells and natural killer T cells are key lymphoid immune cell populations in the liver [23]. NKT cells are developed in the thymus but instead of having properties solely belonging to T cells, they also possess inhibitory receptors of NK cells. They are a key cytokine producer and are heavily involved in hepatic fibrosis through displaying several Toll-like receptors (TLRs). However, NK cell regulation is controlled by multiple stimulatory as well as inhibitory signals to achieve proper function.

In virally infected patients, exposed viral nucleotides, host cell debris, and virus proteins activate different groups of cellular receptors (PRRs), leading to non-specific innate immunity, NK cell proliferation, production of antiviral cytokines, and B & T cell recruitment and maturation. However, HBV may escape immunosurveillance by decreasing its amplification speed and stimulate production of TGF-β and IL-10, while inhibiting the secretion of TNFα and IL-12 induced by TLR2. The high level of IL-10 inhibits the secretion of IFNγ in NK cells, and drives inhibitory receptor PD1 and CD94 expression in NK cells. The sustained binding of PD1/PDL1 and CTLA4/CD94 immune checkpoint proteins keeps stimulating CD8^+^ T cell activity, in turn leading to T cell incompetence [46]. Multiple theories support the notion that in chronic infections, the failure to eliminate viruses is a result of T cell exhaustion. Barber et al. demonstrated via cytotoxic T cell cytokine production and proliferation measurements that although effector T cells are generated in early-stage infection, they slowly lose proper function in lymphocytic choriomeningitis virus (LCMV), human immunodeficiency virus (HIV), HBV, and HCV-infected mice [47]. Consistently high levels of antigens decrease cytotoxic T lymphocyte function and maintain T cell exhaustion, damaging cell proliferation as well as proper transcriptional, epigenetic, and metabolic activities [46].

### 2.4. Alternative Stem Cell Lines for the Treatment of Liver Cirrhosis: A Focus on Mesenchymal Stem Cells

Stem cells have been a focus in tissue engineering because of their unique characteristics that delineate them from other somatic cells. One key feature is the ability to asymmetrically differentiate into a clone, for self-renewal, and a committed progenitor cell [48]. This allows for the maintenance of a steady population of stem cells within the body while also providing a stream of differentiated progenitor cells. While there are a multitude of stem cells within the body, certain types have drawbacks that limit their use in clinically treating cirrhosis. Embryonic stem cells (ESCs) are one such stem cell type. The advantage to ESCs is their capacity to differentiate into any cell type. However, use of ESCs has been met with litigious and political restrictions [49]. Thus, manufacturing a large supply of these cells has been difficult and therefore the therapeutic focus has shifted to substituting the use of ESCs with other cell types.

Great advances were made in 2006 after the discovery of a method to reverse differentiate somatic cells into an induced pluripotent stem cell (iPSC) state uncovered by Kasatochi Takahashi and Shinya Yamanaka [50]. iPSCs have been shown to have the ability to differentiate into any other cell type, similar to that of ESCs. While progress has been made using iPSCs there are some lingering unsolved issues. One challenge is that iPSC-derived hepatocytes have been reported to have an immature phenotype limiting their use for creating/replacing adult hepatocytes [51]. Additionally, cancerous cell lines share a resemblance to gene expression phenotypes found in ESCs, while the processes of dedifferentiating somatic cells into iPSCs exacerbates mutations that lead to oncogenesis and teratomas [52]. These possible tumorigenic characteristics limit ESCs and iPSCs in translational research and thus need to be addressed before their widespread use.

Attention has also been focused on non-stem cell lines such as those found within or near the liver. Certain liver cells have the capacity to differentiate into other types of hepatic cells, such as the hepatoblast. These cells are found in both the fetal and adult liver and can differentiate into cholangiocytes and hepatocytes, the main cell types in the liver. Hepatoblasts are credited for having a leading role in the impressive regenerative potential of the liver after injury. Hepatoblasts have been shown to effectively perform the role that was perceived to be possible only through a resident stem cell niche in the liver, similar to how hematopoietic stem cells are resident stem cells within the bone marrow [53]. Many researchers speculate that the liver does in fact have its own stem cells separate from hepatoblasts. However, there are conflicting reports of the existence of these “hepatic stem cells” and there is much debate whether such cells do in fact exist [54]. The issue centers around the injury state of the outer portal region of the liver lobule, also known as the canals of Hering. The process of regeneration of this region, known as ductular reaction, is mediated by progenitor cells resembling fetal hepatoblasts. The contentions are that these cells only really exist in the fetal liver during epigenesis (80% in fetal vs. 0.01% in adult liver), multiple cell types share the same markers during ductular reaction, making definitive hepatic stem cell identification difficult, and that these cells only express fetal progenitor cell markers during injury [55]. These injury-presenting cells are known as hepatic progenitor cells (HPCs) sometimes called liver progenitor cells (LPCs). HPCs have been reported to differentiate into hepatocytes and cholangiocytes, resembling characteristics similar to that of hepatoblasts with evidence pointing to originating in the canal of Hering [56]. Because HPCs only seem to exist during liver injury of the liver and that hepatoblasts are in low quantities within the adult liver, these cells are not ideal for use in the clinic.

There is therefore a great need for a cellular therapy that can be isolated and expanded at clinically relevant quantities while also alleviating the fibrotic response associated with liver cirrhosis. One cell type that has emerged as a strong candidate is the mesenchymal stem cell (MSC). MSCs refer to cells that show clonogenicity while also being able to differentiate into cells of the mesoderm such as chondrocytes, osteocytes, and adipocytes. They were first discovered in 1961 by Friedenstein and originally isolated from bone marrow (bmMSCs) [57], after which MSCs have been isolated from adipose tissue (adMSCs) [58], umbilical cord (uMSCs) [59], synovial fluid [59] and virtually every organ of the body. Fat and umbilical cords are of special interest since these sources are typically discarded after liposuction and birth allowing for ethnical and easy access to a large source. While having multiple resident tissues may be beneficial for isolation and manufacturing, the origin of isolated cells must be carefully considered when treating patients. In 2006, the international society of cellular therapy published a paper detailing that MSCs should be defined by the markers CD105, CD73, and CD90, and lack expression of CD45, CD34, CD14 or CD11b, CD79a or CD19 and HLA-DR surface molecules [56,60]. This was later shown to not necessarily be true as other cell types were also found to have CD105, CD73, and CD90 as well [61].

Mesenchymal stem cells have slightly different, but significant, phenotypes, depending on where in the body they were isolated [61]. Current strides in single cell sequencing have revealed some insights into the subsets of MSCs. In a recent report, 11 MSC subsets were identified only in the umbilical cord and synovial fluid [59]. One study found that adMSCs more effectively inhibit the differentiation of monocytes to dendritic cells compared to bmMSCs [58]. In fact, many papers cite uMSCs as the most immunomodulatory [62]. One study found that uMSCs released IL-6, MIP2, and GRO more than adMSCs, with IL-6 and MIP2 being chemoattractants for leukocytes [63]. Furthermore, one study comparing the three primary locations (bone marrow, fat, and umbilical cord) found that mesenchymal stem cells derived from umbilical cords had the highest proliferation rate and highest immune inhibitory effect when cocultured with macrophages [64]. Additionally, this distinction includes differentiating potential. A paper published by Lee et al. outlines a procedure to differentiate bmMSCs and uMSCs into hepatocytes in serum-reduced conditions with exposure to hepatocyte growth factor, basic fibroblast growth factor (bFGF), oncostatin M, dexamethasone and cell culture supplement ITS+ (insulin, human transferrin, selenous acid). The researchers demonstrated that the newly differentiated cells were able to produce albumin, store glycogen, and secrete urea after 12 weeks as well as demonstrate a cuboidal and shortened morphology [65]. These key functions are indicative of hepatocytes and are not at all or scarcely observed with undifferentiated MSCs. One study found that MSCs show a chimeric phenotype for mesenchyme and hepatocytes when in vitro, yet in vivo MSCs completely lose this chimeric phenotype and fully differentiate into hepatocytes [66]. Kazemnejad et al. reported seeding bmMSCs onto an electrospun PCL/Collagen/PES nanofiber polymer slab and subsequently differentiating the bmMSCs using the technique outlined above. The study revealed the production of albumin increased from 28 ± 6% in 2D culture to 47 ± 4% while on the nanofiber slab, and further highlights the benefit of including ECM proteins such as collagen while also implementing nanofiber structure to recapitulate native ECM morphology [67]. Lack of human serum albumin can lead to an increase in oxidative molecules in the blood and lower transport efficiency, considered to be a distinguished sign of liver dysfunction [68].

While differentiating stem cells can help replace lost hepatocytes, it has been shown that the primary role of MSCs in an injury model is through immune modulation rather than differentiation [69,70]. In one report, MSCs were shown to release IL-6 and HGF to inhibit the proliferation of NK cells, macrophages, and monocytes [71]. Another study outlines an MSC function as a target for apoptosis mediated by cytotoxic T cell and, upon rupture, release their immuno-modulatory cytokines into the microenvironment mediated by HpSCs [72,73].

## 3. Liver Fibrosis Therapies

Liver fibrosis and cirrhosis can be detrimental, given that the liver is the site of blood filtration with multiple immune populations. The liver can regenerate itself but not on a large scale, requiring treatment and therapeutic intervention to mediate developmental defects, genetic deficiency, liver wound healing, liver transplants, and diseases such as hepatitis B and C, cancer, alcoholic steatohepatitis (ASH), alcoholic fatty acid disease (AFAD), non-alcoholic steatohepatitis (NASH), non-alcoholic fatty acid disease (NAFAD), autoimmunity, chemical exposure, and pathogens. This section aims to discuss the recent progress completed in both preclinical and clinical realms relating to interventions for liver fibrosis. These methods range from mesenchymal stem cells, gene delivery, monoclonal antibodies, and targeting of molecular pathways.

### 3.1. Pre-Clinical Liver Fibrosis Treatments with MSCs

In pre-clinical research, liver fibrosis is typically induced in animal models using subcutaneous or intraperitoneal injections of carbon tetrachloride (CCl4). Molecular mechanisms, changes in biological processes, and pathways in toxin-induced fibrosis were investigated by one group via transcriptomic and proteomic analyses [74]. From the differentially expressed gene and protein groups, they identified 523 overlapping proteins from both transcriptomic and proteomic analyses involved in processes such as response to oxidative stress, cellular response to extracellular stimulus, and extracellular matrix/structure and actin filament reorganization. Therefore, MSCs are a viable option for cirrhosis treatment given their ability to produce various immunomodulatory soluble factors.

Zhao et al. describes a study conducted to evaluate the use of MSC administration [75]. Rat MSCs were isolated from bone marrow and induced to differentiate into hepatocytes ex vivo using HGF, FGF-4, and epidermal growth factor (EGF) then labeled with DAPI for in vivo tracking. The labeled MSCs were prepared at 10^7^ cells/mL and 300 uL was injected as an intraperitoneal (IP), intravenous (IV), or intrahepatic transplantation 28 days before sacrifice. Molecular and biochemical assays such as analysis of albumin, total bilirubin in serum (TBIL), and ALT revealed IV to be the most favorable mode of administration, given that it yielded the highest levels of surviving homing labeled MSCs, normal liver lobe morphology, and minimal collagen deposition when compared to the control with no treatment. Histology revealed reduced α-SMA in the IV administration group and the serum levels of alpha-fetoprotein (AFP), albumin, and TBIL were close to normal, compared to IP and intrahepatic groups. Both qPCR and ELISA analysis confirmed enrichment of IL-10, an anti-inflammatory cytokine produced by T helper 1 (Th1) cells balancing MMP activity from activated myofibroblasts thus preventing fibrosis, in the IV group [76]. This study also indicated the importance of the administration method for both pre-clinical and clinical studies.

Similarly, another study examines the intravenous administration of autologous bmMSCs confirmed by CD29 gene expression, but omitted the ex vivo differentiation step; instead, the researchers injected rats with 3 × 10^6^ cells and were maintained for 28 days [77]. They noted elevated serum albumin levels, indicating liver function and significantly lower collagen accumulation via histological observations, decreased expression of *Col1a*, and significantly reduced hydroxyproline content, an indicator for the amount of collagen. These studies highlight the benefit for IV injection of MSCs. However, other exciting work is being conducted with varying transplant methods.

Investigators explored the microencapsulation of human adult bmMSCs to further analyze the mechanisms governing MSC therapy in the context of liver fibrosis (Figure 2) [78]. The microspheres are composed of an alginate-polyethylene glycol blend, allowing the cells to proliferate, differentiate, and allow for the diffusion of soluble factors—such as cytokines IL-6, IGFBP-2, which aids MSC differentiation and self-renewal, and MCP-1, in and out of the sphere while protecting the xenogeneic MSCs from immune-mediated rejection. These microspheres were injected intraperitoneally in two different chronic liver fibrosis mouse models—toxicity CCl4-induced and bile duct ligation (BDL) injury. After 15 days, analysis was conducted on tissues and blood samples revealing lower collagen type 1 and α-SMA expression, higher levels of endogenous IL-10 and MMP-9 activity, indicating matrix degradation and fibrosis resolution. Detection of human IL-1Ra in those transplanted with microencapsulated MSCs indicate inhibition of inflammation. This study is unique in that the immune protective characteristic of the microspheres allows for the function and mechanistic elucidation of human adult bmMSCs.

The use of exosomes is a favored technique when it comes to fibrotic liver mouse models. Exosomes allow MSCs to communicate with nearby cells and include a snapshot of cellular activity via proteins, RNAs, and metabolites [32]. A study was conducted with CCl4-induced mouse models and human-derived MSC exosomes. The exosomes were injected directly into the left and right lobes of the liver. Serum levels of TGF-β1 decreased after treatment and thus inhibited the transforming growth factor/Smad pathway that allows for epithelial-mesenchymal transition (EMT) of liver cells. Immunohistochemistry analysis showed that after 1 week of treatment there was a visible reduction in N-cadherin, collagen deposition, and vimentin-positive cells. Further, an in vitro experiment was carried out where the hepatic cell line HL7702 was treated with hTGF-β1 to transform them to fibroblasts via EMT, then 100 ug/mL huMSC-Ex were added. The fibroblasts reverted back to a native spindle-shaped morphology in addition to a decreased expression of N-cadherin, showing promise in alleviating hepatic inflammation. In a separate study using a similar cirrhotic mouse model, researchers injected mice intravenously with MSCs and reported decreases in gene expression of *TGF-β*, alpha Smooth Muscle actin (α*-SMA*), collagen 1 (*Col1a*) and *TNF*α after injecting CCL-4 induced cirrhotic mice with MSCs. The results of these immune inhibitory mechanisms range from decreased weight loss to a reduction in fibrotic ECM with cellular intravenous injection [79] (Figure 3). While these studies show promising results, rat and mouse models are not directly equivalent to human trials. For example, murine MSCs become cancerous after a shorter passage number than their human counterparts [61]. Efforts have been made to test the efficacy of human MSC therapy models in non-human primates (NHPs). One study found that biweekly intravenous injection of human MSCs into Cynomolgus macaques at an infusion rate of 3–4 million cells/min showed no significant increase in immune cell markers in peripheral blood over controls [80]. Another larger scale study on acute liver failure found that Rhesus macaques that were given peripheral human MSCs showed significant decreases in IL-6 and IL-15, both of which have been associated with pro-inflammatory cytokine storm [81]. Therefore, the use of the immunomodulatory effects of MSCs should be further expanded to investigate their effects in treating fibrotic liver damage.

Other groups have applied the concept of decellularized liver scaffolds for efficient MSC differentiation and then inject them into fibrotic liver mouse models. Instead of seeding hepatocytes, murine MSCs were seeded within the decellularized rat liver 3D matrix, which supported their differentiation and maturation into hepatocytes [82]. Introduction of growth factors (GF) increased the differentiation efficiency by 24.5% in the dynamic scaffold compared to a tissue culture flask. The MSCs within the scaffold displayed hepatocyte structural changes and expressed hepatocyte-related genes for a-1-antitrpsin, transthyretin, and glucose-6-phosphatase. MSCs from the dynamic scaffold were then injected via tail vein into a CCl4-induced fibrotic mouse model. MSCs differentiated in the dynamic cell scaffold established an increase in function, differentiation, and cell survival compared to MSCs differentiated in tissue culture flasks or treated with GF. The gene expression levels of *α**-SMa* decreased in the mice treated with dynamic scaffold-differentiated MSCs compared to flask cultured MSCs, which indicates a decrease in activation of HpSCs likely responsible for the morphological fibrotic changes.

### 3.2. Clinical Translation of Liver Fibrosis Therapies with MSCs

Within the past decade, many exciting translational efforts in the context of liver fibrosis treatment and diagnosis have been initiated and are now at various clinical trial stages. Table 1 highlights trials with great relevancy, presenting significant improvements within the field. The clinical landscape of fibrosis reversal or alleviation spans the use of different targets and methods, such as MSCs, small molecules, genetic modifications, delivery methods, monoclonal antibodies, and diagnosis tools.

The manipulation and use of MSCs dominate clinical developments relating to antifibrotic treatments for the liver. For instance, a clinical trial with Phases 1 and 2 investigates the differentiation of autologous MSC into hepatocyte progenitors. These progenitor cells are then injected using ultrasound guidance into the patient via the portal vein to treat liver cirrhosis and failure resulting from different diseases, including cryptogenic, hepatitis B, C, and alcoholic hepatitis [83]. The researchers reported injections to be well tolerated within all patients, improved scores based on the Model for End-Stage Liver Disease (MELD) rubric, and decreased creatinine and prothrombin complexation indicating improved liver function [83]. The autologous MSC treatment did not change serum bilirubin and only marginally increased serum albumin. Alternatively, another group harnessed MSC therapy combination instead with 30 mg/day of pioglitazone, a peroxisome proliferator-activated receptor-γ (PPAR-γ) agonist, to treat liver fibrosis in a Phase I trial [84]. PPAR-γ has been shown in vitro to have an influence on the remodeling of extracellular matrix and thus decreases liver fibrosis levels [85]. Therefore, activation of PPAR-γ may have implications in reducing the activation of HpSCs into MFs and thus reducing expression of α-smooth muscle actin and collagen type 1 expression. Patients were treated twice within 6 months, with the therapy being well tolerated with no signs of further deterioration, and improved MELD scores were reported after 3 months. While there were only two patients enrolled, the study showed safety and tolerability. Separate work has also been done with bmMSCs. A group in China conducted Phase II and III clinical trials analyzing the use of autologous bmMSCs to treat HBV-induced cirrhosis, but more specifically to analyze the regulation of Treg and helper Th17 cells post-transplantation [86]. A total of 36 of the 56 enrolled patients completed the study, with all patients receiving doses of Entecavir. Those also treated with MSC transplantations presented a decrease in Th17 cells and thus an increase in Treg cell populations, overall increasing the Treg/Th17 ratio. The same changes were shown at the mRNA expression level with markers *Foxp3* and *ROR**γτ*, respectively, alongside a decrease in inflammatory factors such as *IL**-6*, *TNF**α*, and *IL**-17*. The authors concluded that due to the regulation of the Treg/Th17 ratio, liver function and fibrosis was improved in the treatment group [87]. Another clinical trial investigated the use of bmMSCs along with antivirals for the treatment of liver cirrhosis as a result of hepatitis B, but results are not posted at this time [88].

Other scientists and clinicians have explored regeneration of liver tissue via autologous adipose-derived stromal MSCs [86]. Instead of portal vein injection, the MSCs were introduced into patients with intrahepatic arterial administration. The one-month study spanned three different types of liver disease and no serious adverse effects were observed following cell therapy infusion [89]. Serum albumin levels were improved and maintained in three of the four patients when clinicians followed up over six months to one year later. Factors such as IL-6, HGF, and macrophage inflammatory protein 1-beta increased after 24 h, indicating liver regeneration.

While these studies and trials show promise, it is also important to keep in mind the potential downsides of MSC therapy. Many clinical trials have faced issues with efficacy even when safety was demonstrated, and the cells used for these therapies were not always entirely characterized [78]. Clinical trial studies must be presented with compatible controls and design, such as randomizations, indicating that much clinical work is necessary before such therapies can reach later phases.

While cellular therapies may allow for autologous derivation and therefore bypass transplant complications and waitlists, they are largely aimed at improving tissue regeneration. In a particular Phase I/II study, expanded autologous CD34^+^ hematopoietic stem cells were manipulated and used to treat patients with chronic liver insufficiency [90]. The recently completed safety and tolerability trial enlisted five patients and identified cell doses starting at 1 × 10^9^ with a maximum of 5 × 10^9^ cells infused through the hepatic artery or portal vein. In terms of responses, one patient experienced a urinary tract infection that required hospitalization and an antibiotic prescription, four reported pain and fever, while all five reported nausea. Although patients did not experience adverse effects that cause termination of the trial, it is important to note that cell therapies can cause systemic effects and symptoms. Mesenchymal stem cells are also manipulated ex vivo for transplant and therapy, including those for the treatment of hepatitis C cirrhosis. In one Phase I/II study, adipose-derived MSCs were expanded ex vivo, then transplanted into patients experiencing liver cirrhosis at 1 × 10^6^ cells/kg via the peripheral vein or 3 × 10^6^ cells/kg via the hepatic artery three times every two weeks [91]. The cell-based therapy was well tolerated in the 25 patients enlisted, during which 13 patients were excluded due to liver transplants, death, or inability to expand MSCs and pass quality control assessments, leaving 12 patients following through to the end of the study. Improvement in MELD scores was noted in 8 of the 25 patients and albumin levels from serum testing increased after 3 months. Although five patients reported a lower hepatitis activity index score, analysis of the liver via biopsy revealed no significant differences in tissue regeneration, which may indicate that the bioavailability of MSCs in the lesion area was insufficient [92]. Interestingly, viral HCV RNA levels decreased to complete clearance, which was predicted to be a result of potential endocrine immunomodulatory mediators produced by the MSCs. Cell therapies are a promising alternative treatment to transplantation and encouraging results are observed from the use of stem cells. Although, setbacks and limitations include an insufficient amount of tissue to derive the cells of interest and/or low quality of donor preps post-isolation.

### 3.3. Gene Delivery and Genetically Modified Stem Cells

To further the applications of MSCs as a therapy, genetic modification allows for improved homing capabilities. MSCs travel to the site of tissue damage through a series of receptors and ligand interactions, such as CD44 and P selectins; CXCR4 and SDF-1; or VLA-4 and VCAM-1. Once the MSCs engage with endothelial cells, remodeling enzyme MMPs are secreted by the stem cells and degrade the basement membrane of the endothelium [93]. Marquez-Curtis et al. describes a procedure in which MSCs are genetically modified to express high levels of CXCR4 [94]. The group reports a 10^5^-fold higher expression level of CXCR4 and a 3-fold increased rate of migration towards SDF-1 gradients, while maintaining differentiation abilities. This transfection technique could be applied to MSCs with the aim of improving homing efficiency to liver lesions.

Genetic modification extends to other cell types as well, in the form of gene silencing. One study investigated silencing TGF-β1 in a hepatic stellate cell HpSC-T6 cell line [95]. The silencing was completed using short hairpin RNA (shRNA) and small interfering RNA (siRNA) separately via pyridinium lipid/l-α-dioleoyl phosphatidylethanolamine (DOPE) cationic liposomal and Lipofectamine transfection with qPCR and Western blot analysis. Silencing TGF-β1 in turn allows for the increased degradation of extracellular matrix by preventing the downregulation of matrix-degrading enzymes and TIMP-1 and thus decrease in α-SMactin and collagen type 1 presence. In addition to the presence of extracellular matrix markers, the authors reported a decrease in inflammatory cytokines including IL-1β and TNFα. These results suggest that genetic silencing of TGF-β1 in HpSCs can have beneficial effects in terms of decreased collagen deposition and inflammation, thus alleviating liver fibrosis pathology.

Furthermore, microRNAs (miRNAs) have been used in treating liver pathogenesis. These small non-coding RNAs can regulate the expression of certain genes by targeting mRNAs. Many groups are researching the presence of miRNAs, mainly the miR-29 family, and the relationship with liver fibrosis [96]. Within the miR-29 family, miR-29b is of interest for its ability to target genes related to liver fibrosis such as protein kinase B (PKB) also known as AKT, collagen type 1, and platelet derived growth factor-beta (PDGFβ). Kumar et al. describes the development of a cationic copolymer micellular delivery system to treat C57BL/6J male mice loaded with both miR-29b and GDC-0449, a hedgehog (Hh) inhibitor that decreases the number of myofibroblasts in the hepatic system. Co-encapsulation ensures stability of the miRNA and solubility of GDC-0449 while maintaining similar bioavailability. Intravenous injection into a CBDL-induced liver fibrosis model showed a decrease in inflammation, collagen deposition in the liver, infarctions, and a decrease in protein markers for phosphorylated AKT and PDGFβ, suggesting the potential of miRNA treatment as a promising therapy.

Other miRNA genes have been investigated in the context of liver fibrosis, including miR-378a-3p [97]. This miRNA molecule targets the glioblastoma family protein 3 (Gli3) within HpSCs and thus suppresses their activation. When phosphorylated, Gli3 enters the nucleus and initiates Hh signaling. Encapsulated miR-378a-3p within l-tyrosine polyurethane, PEG, and hexamethylene diisocyanate-formulated nanoparticles were injected intraperitoneally into CCl4-treated C57BL/6J male mice. The authors identified a decrease in *Gli3* and matrix marker (*α**-SMa*, *Col1a1*, and *Timp1)* expression, normal ALT and AST levels due to normalized Hh signaling, and morphology resembling healthy control histology.

Beyond miRNA technology, the gene delivery field has also harnessed the use of viral delivery for treating liver fibrosis. Zhong et al. describes an adenoviral delivery system for the enzyme superoxide dismutase (SOD), which is known to minimize oxidative stress that results from cholestasis-induced hepatic injury [98]. The Ad-Mn-SOD model, when delivered a few days before common bile duct ligation (CBDL), revealed a decrease in *Col1a1* expression, TGF-β and TNFα synthesis, and focal necrosis pathology. The authors claim the delivery of Ad-Mn-SOD can prevent damage by inhibiting toxic cytokine and free oxygen radical generation that usually results from cholestasis. Therefore, this shows that gene delivery by viral vectors may also be clinically relevant for preventing further liver decomposition in at risk patients.

### 3.4. Targeted Molecular Pathways and Small Molecule Drug Delivery

Interestingly, researchers have also investigated unique antagonists to halt the progression of fibrosis in the liver. One group explored the activation of cannabinoid CB1 receptors in the liver, given it had been previously shown that CB2 receptor activation influences both CB2-independent profibrogenic and CB2-mediated antifibrogenic effects [112]. Western blot analysis revealed high levels of CB1 in fibrotic models compared to wild type mice. The receptor antagonist for CB1 is called SR141716A, which decreases the induction of TGF-β1 and α-SMa expression and thus inhibits the increase of myofibroblasts and fibrogenic hepatic cells triggered by cell death. Further efforts are necessary to elucidate the CB1 receptor activation mechanism prior to becoming clinically relevant.

In terms of small molecules, an effective drug with the ability to reverse liver fibrosis does not yet exist. On group investigated the targeted delivery of an angiotensin II type 1 (AT1) receptor blocker conjugate to HpSCs to increase the uptake of the drug and efficacy in eliminating fibrotic liver tissues. The molecule used is called losartan-M6PHSA, a linked AT1 receptor blocker with a HpSC-selective drug carrier mannose-6-phosphate-modified human serum albumin [113]. BDL- and CCl4-induced rat models were treated with losartan-M6PHSA and compared to losartan or M6PHSA alone. The authors carried out computer-based morphometric quantification to analyze the presence of specific cells or molecules, such as myofibroblasts, CD43^+^ inflammatory cells, and collagen deposits. Immunostaining revealed the successful co-localization of losartan-M6PHSA with HpSCs with reduced collagen accumulation, myofibroblast presence, and the mRNA expression of the procollagen α2(I) (*Col1a2*). Terminal deoxynucleotidyl transferase-mediated deoxyuridine triphosphate nick-end labeling (TUNEL) was used to identify apoptotic cells and revealed that losartan-M6PHSA did not have an association with the apoptosis of activated HpSCs. In addition to this, the drug conjugate did not show an effect on the activity of myofibroblast collagenolytic enzymes MMP2/9.

Clinically, angiotensin receptor blockers are used to ease vessels to alleviate and lower portal pressure, but in a diseased liver the blocking of this pathway can prevent stellate cell stimulation and myofibroblast activation. This can lead to fibrogenesis and thus have anti-fibrotic effects [114]. Oral losartan is an angiotensin II type I (AT1) antagonist used in clinical trials to reduce fibrosis inflammation hepatitis C patients [115]. Long-term administration of losartan at 50 mg per day for 18 months in 14 patients was then analyzed for collagen content and expression of specific genes in a Phase 4 clinical trial [106]. Although well tolerated in all patients, the serum levels of AST and albumin did not change after oral losartan. The treatment resulted in stable collagen content in all patients, but the degree of fibrosis decreased in half. Gene expression analysis via qPCR revealed a decrease in genes for proteins such as urokinase-type plasminogen activator, ras-related C3 botulinum toxin substrate 1, NOX activator 1, NOX organizer 1, procollagen α1(I) and α1(IV), which are identified as profibrogenic. The decrease in these genetic profiles indicates inhibition of collagen deposition in addition to preventing oxidative stress, thus attenuating fibrous scarring. Further, urokinase-type plasminogen activator is involved in the extracellular matrix regeneration cycle, and therefore decreased levels assist excessive inflammation and angiogenesis, which leads to the pathogenesis of fibrosis. The study showed safety, specifically lack of renal impairment, and downregulation of fibrogenic factors in chronic hepatic C patients, but further work is necessary to evaluate its effect in other fibrosis models. Irbesartan is another small molecule drug that targets the same angiotensin system for use in treating fibrosis. In a Phase 3 clinical trial, irbesartan was administered in pill form at 150 mg per day for 2 years [116]. As an AT1 receptor antagonist, irbesartan has the potential to partially or completely inhibit TGFβ levels that have a role in fibrotic formation in the liver.

Additionally, small molecule pirfenidone, also known as 5-methyl-1-phenyl-2-(1H)-pyridone, has been used to inhibit collagen generation. A Phase II study investigated the use of this antifibrotic drug taken orally in prolonged-released form at 600 mg every 12 h for advanced hepatic fibrosis. A total of 281 participants of varying chronic liver disease etiology, including hepatitis C, fatty liver disease, and alcohol-induced fibrosis, were enrolled and 122 of those were administered the regime for 12 months [117]. Current results reported in Poo et al. describe decreased fibrosis in 35% of patients administered prolonged-release pirfenidone compared to only 4% in those administered non-prolonged-release pirfenidone [117]. Interestingly, ALT and AST serum levels decreased similarly for both regimes around 40–43%. Patients treated with pirfenidone regimes experienced lower levels of TGF-β1, IL-6, and endothelin-1 compared to control patients receiving standard of care (nutritional support, annual upper-gastrointestinal endoscopy, bi-annual liver ultrasound, etc.). These changes at the cytokine level had effects on anti-fibrotic progression by preventing HpSC stimulation and thus transition to myofibroblasts. This study suggests the safety of pirfenidone and the efficacy of prolonged release compared to the standard of care alone in addition to improving fibrotic characteristics such as stiffness and inflammation. It is important to note that most of these trials have small patient sample sizes and varied results. Therefore, further elucidation of the mechanisms and dosing schemes is required.

### 3.5. Monoclonal Antibodies

Monoclonal antibodies have also been investigated for their application in alleviating liver fibrosis. Ogawa et al. evaluated the efficacy of a monoclonal antibody specific for platelet derived growth factor polypeptide B chain (PDGF-B) in reducing liver fibrosis [118]. PDGF is involved in the chemotaxis of MFs and HpSCs, contributing to the development of fibrotic pathology. The antibody developed, AbyD3263, prevents phosphorylation of receptors involved in the downstream ERK pathway. Blocking the ERK pathway halts the activation of HpSCs. After neuralization and inhibition of PDGFRβ phosphorylation was confirmed, AbyD3263 was injected into both BDL- and concanavalin A-induced fibrosis models alongside a control antibody, imatinib, which blocks both PDGFRα and PDGFRβ in BDL-induced mice. Imatinib decreased hydroxyproline levels by 58.9% whereas AbyD3263 decreased hydroxyproline levels by 38.7% compared to the control. In concanavalin A acute liver injury models, AbyD3263 did not eliminate necrotic areas but reduced fibrosis. While further studies will need to be conducted to elucidate the AbyD3263 dosage needed for target neutralization, these initial findings suggest AbyD3263 as a promising therapy.

A recently concluded Phase 2 clinical trial investigated the use of a monoclonal antibody, simtuzumab, to treat liver fibrosis due to NASH [119]. The antibody is designed against lysyl oxidase-like molecule 2, which initiates linking between collagen and thus leads to fibrogenesis [119]. The study was concluded after 86 weeks due to a lack of efficacy in decreasing the fibrosis stage and therefore allowing progression to serious cirrhosis. There was not a statistically significant decrease in hepatic collagen content between patients receiving simtuzumab at 125 mg via subcutaneous injection once a week compared to the placebo group. Another study accesses the use of monoclonal antibodies against TNFα in 12 patients with severe alcoholic hepatitis [120]. A single injection of infliximab at 5 mg/kg was administered to patients and samples were collected to analyze cytokine and serum levels. Within the first 30 days, levels of serum bilirubin, c-reactive protein, and neutrophil count decreased, but unfortunately, two patients died due to septicemia. Interestingly, the mRNA expression levels of TNFα did not change due to the treatment but expression levels of cytokine IL-8, which is under the direct influence of TNFα, were undetectable. More recently, a clinical trial posted in February 2021 is currently recruiting patient participants with ALD to determine the safety and tolerability of Guselkumab, a humanized monoclonal antibody for IL-23 previously approved for use in psoriatic arthritis [107,121]. These initial studies highlight the need for more localized release therapies to mitigate complications associated with systemic antibody delivery treatments, such as septicemia.

## 4. Conclusions

Various diseases cause prolonged inflammation of liver tissue, which has the potential to develop into cirrhosis. Understanding the specific immune mechanisms related with each outlined disease is crucial in halting the progression of cirrhosis and aiding in reversing fibrotic effects. Mesenchymal stem cells, through their multiple avenues of isolation, are an abundant source of cells. Efforts have been made to utilize the differentiation potential of MSCs to replace damaged hepatocytes. Importantly, MSCs have immunomodulatory effects in promoting activation of T-regs and inhibiting proliferation of cell populations such as NK cells and macrophages via cell-cell contact and secreted cytokines. They have been shown to deactivate hepatic stellate cells and have been shown in murine models to reduce fibrotic progression of cirrhosis. Various treatment methods have been explored spanning from autologous MSC transplants, molecular pathway targeting small molecules, gene delivery and cell modification, scaffold engineering, and monoclonal antibodies. Recent pre-clinical and clinical advances towards both liver fibrosis treatment and diagnosis devices hold promise in warranting the continuation and furthering of these studies.

## Figures and Tables

**Figure 1 ijms-22-06777-f001:**
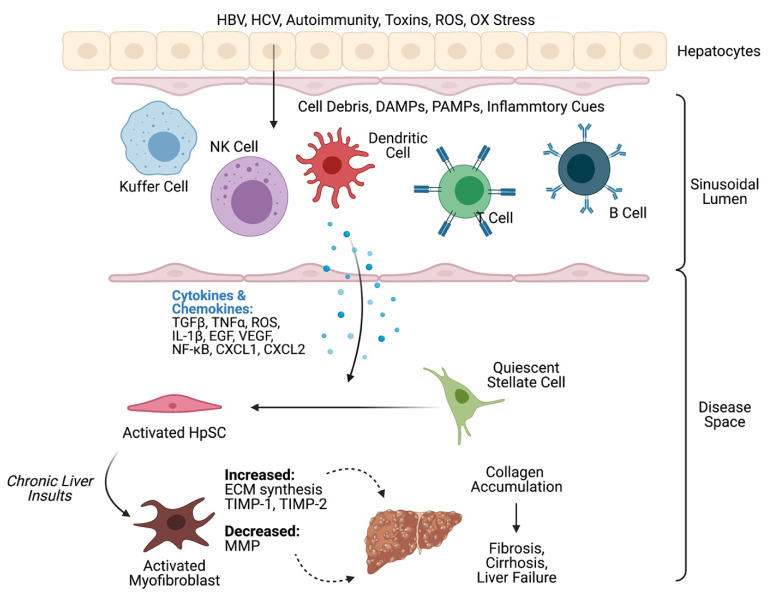
Original schematic depicting the immunological mechanism behind liver fibrosis, cirrhosis, and failure. A summary schematic of hepatic stellate cell (HpSC)-initiated extracellular matrxi (ECM) deposition, where imbalanced tissue inhibitors of metalloproteinase (TIMP)/matrix metalloproteinase (MMP) ratios lead to less proteolytic activity and increased collagen synthesis. HBV: hepatitis B virus, HCV: hepatitis C virus, DAMP: damage-associated molecular pattern, PAMP: pathogen-associated molecular pattern, NK: natural killer, TGF-β: transforming growth factor β, TNFα: tumor necrosis factor-alpha, ROS: reactive oxygen species, IL: interleukin, EGF: epidermal growth factor, VEGF: vascular endothelial growth factor, NF-κB: nuclear factor-kappa B, CXCL: chemokine (C-X-C motif) ligand. Original schematic was created with BioRender.com.

**Figure 2 ijms-22-06777-f002:**
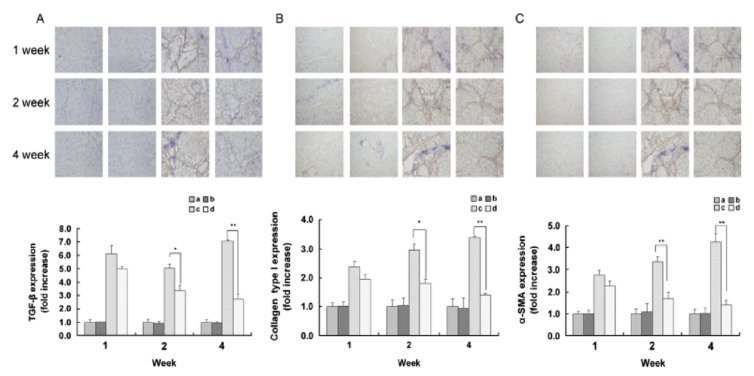
Expression of select fibrotic factors via immunohistochemistry. (**A**–**C**). TGF-β1 (**A**), collagen type 1 (**B**) and α-smooth muscle actin (**C**) expression at 1, 2, and 4 weeks. It can be visually determined that the fibrotic effect is reduced with treatment of human umbilical cord blood-derived mesenchymal stem cells (hMSCs) as opposed to CCl4 induced cirrhosis and saline. Mice livers injected with saline (left), olive oil and hMSC injection (left center), CCl4 induced cirrhosis with saline injection (right center) and CCl4 induced cirrhosis with hMSC injection (right) on each of the three panels. The MSC injection group shows a 4-fold decrease in αSMA and TGF-β and a 3-fold decrease in collagen 1 at 4 weeks with *n* = 5 mice. The study demonstrates MSC homing into the cirrhotic livers more so compared to healthy livers, further signifying the efficacy of the cellular therapy. Original magnification of 200×. Values are presented as mean ± SEM, for at least five separate experiments. Data are presented as fold change with respect to saline/CCl4 group, ***
*p* < 0.05, ** *p* < 0.01. Reprinted with permissions from Jung et al. [79].

**Figure 3 ijms-22-06777-f003:**
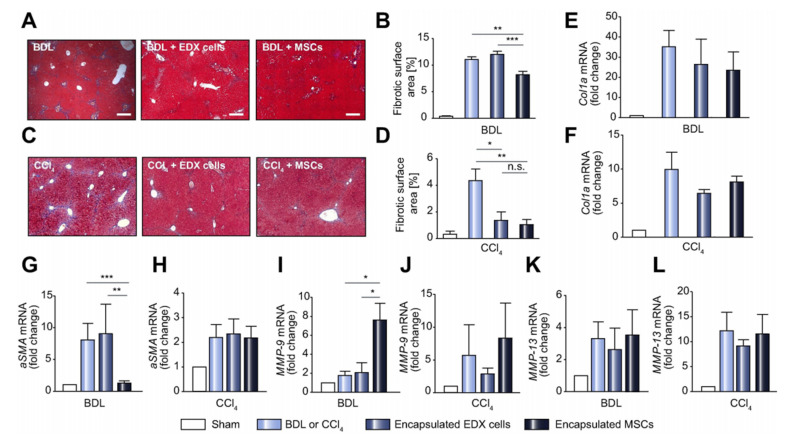
Transplanted microencapsulated mesenchymal stem cells can be used to aid liver fibrosis. (**A**–**D**) MSCs were encapsulated in novel alginate-polyethylene glycol microspheres and transplanted into BDL or CCl4-induced mice, then livers were harvested after 15 days or 4 weeks, respectively. (**A**,**C**) Morphometric quantification of Masson’s trichrome stained sections was calculated, showing a decrease in fibrotic surface area with encapsulated MSC treatment. Scale bars are 400 μm. (**E**–**L**) Real time-PCR analysis was conducted for *Col1a*, *α**-SMa*, *MMP-9*, and *MMP-13* gene markers. The results reveal reduced expression of collagen type 1 in with encapsulated MSC treatment compared to untreated. Increase in *α**-SMa*, *MMP-9*, and *MMP-13* gene expression was observed in both BDL and CCl4-induced models when treated with encapsulated MSCs compared to encapsulated foreskin fibroblasts. MMP-9 is known to be overexpressed in cell population such as neutrophils and lymphocytes, which can curb accumulation of extracellular matrix and thus reduce liver fibrosis pathology. Data presented as fold change with respect to housekeeping genes and expressed as mean value ± SEM. * *p* < 0.05, ** *p* < 0.01, *** *p* < 0.001. Reprinted with permissions from Meier et al. [78].

**Table 1 ijms-22-06777-t001:** Selected clinical treatments, therapies, and diagnostic tools for liver fibrosis.

General Category	Target Disease Condition	Therapeutic Agent	Responsible Company/Lab	Phase	ClinicalTrials.gov Identifier
Stem Cells	Compensated Liver Cirrhosis	Autologous Mesenchymal stem cells (MSCs) and Pioglitazone	Royan Institute	Phase I	NCT01454336 [84]
	Hepatitis C Virus (HCV) Infection	Autologous Mesenchymal stem cells (MSCs)	Saglik Bilimleri Universitesi Gulhane Tip Fakultesi	Phase I/II	NCT02705742 [91]
	End-stage Liver Cirrhosis	Autograft MSCs Differentiated Into Progenitor of Hepatocytes	Shahid Beheshti University of Medical Sciences and Tarbiat Modarres University	Phase I/II	NCT00420134 [99]
	Chronic liver insufficiency	Autologous Expanded CD34^+^ Haemopoietic cells	Imperial College London	Phase I/II	NCT00655707 [90]
	Liver Cirrhosis	Co-transferring of MSCs and Tregs	Nanjing Medical University	Phase I/II	NCT03460795 [100]
	Liver Cirrhosis	Autologous Bone Mesenchymal Stem Cells (bMSCs) via Portal Vein	Sun Yat-sen University	Phase II	NCT00993941 [88]
	Acute-On-Chronic Liver Failure	Human umbilical cord-derived mesenchyme stem cells (hUC-MSCs)	Hai Li, Shanghai Jiao Tong University School of Medicine	Phase II	NCT04822922 [101]
	Liver Cirrhosis, portal hypertension, hepatic Decompensation	Autologous bone marrow stem cells infusion (ABMSCi) and abdominal portal hypertension surgery	Wenzhou Medical University	Phase II/III	NCT01560845 [86]
	Liver Cirrhosis	Intrahepatic Arterial Administration of Autologous Adipose Tissue Derived Stromal Cells	Kanazawa University	N/A	NCT01062750 [102]
Small Molecules	Chronic Hepatitis C Virus (HCV)	Small Molecule Agent (PF-868554), direct antiviral agent	Pfizer	Phase I	NCT00671671 [17]
	Chronic Hepatitis C Virus (HCV) Infection	Emricasan (IDN-6556)	Conatus Pharmaceuticals Inc.	Phase II	NCT02138253 [103]
	Chronic Liver Fibrosis	Prolonged-Release Pirfenidone Formulation	Grupo Mexicano para el Estudios de las Enfermedades Hepaticas	Phase II	NCT04099407 [104]
	Hepatic Fibrosis in Chronic Hepatitis C	Irbesartan	French National Institute for Health and Medical Research-French National Agency for Research on AIDS and Viral Hepatitis (Inserm-ANRS) and Sanofi	Phase III	NCT00265642 [105]
	Hepatic Fibrogenesis in Chronic Hepatitis C	Losartan	Hospital Clinic of Barcelona	Phase IV	NCT00298714 [106]
Antibody Therapy	Alcoholic Liver Disease	Guselkumab, a humanized anti-IL23 monoclonal antibody	University of California, San Diego	Phase I	NCT04736966 [107]
	Non-alcoholic Fatty Acid Liver Disease	IMM-124E, polyclonal antibody against endotoxin lipopolysaccharide (LPS)	Immuron Ltd., Emory University, and Advanced MR Analytics AB	Phase II	NCT03042767 [108]
Diagnostic Tools	Liver Fibrosis and Congestion in Fontan Patients	Non-Contrast Magnetic Resonance Imaging, Device	Children’s Hospital Medical Center, Cincinnati	N/A	NCT03539757 [109]
	Liver Fibrosis	Mechanical Vibrations with Ultrasound Shear Wave Imaging, Device	Mayo Clinic	N/A	NCT03637959 [110]
	Liver Fibrosis	Fibroscan^®^ of Echosens, Aixplorer^®^ of Supersonic Imagine, Aplio XG of Toshiba, QRS software developed by Pr I.Bricault, Acuson S2000 of Siemens, Device	University Hospital, Grenoble & Clinical Investigation Centre for Innovative Technology Network	N/A	NCT01537965 [111]

## Data Availability

The clinical trial data presented in this review are openly available at clinicaltrials.gov.

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
