# Peer review of "Liver Disease: Induction, Progression, Immunological Mechanisms, and Therapeutic Interventions"

_ijms, 2021, doi:10.3390/ijms22136777_

Round 1
Reviewer 1 Report
The paper described various aspects of liver diseases, mainly focusing on the treatment to improve the advanced liver fibrosis. Their descriptions in the basic sciences, particularly cellular transplantations, may provide some beneficial information for readers. However, their descriptions regarding the clinical studies seem to include some too obsolete data.
1) Section 2.3 Chronic Hepatitis C: Interferon-based treatments for chronic hepatitis C is not used in the current clinical practice. Major guideline unanimously recommend the treatments with direct antiviral agents/direct-acting antivirals (DAAs).
2) Table 1: As they mentioned, clinicaltrials.gov NCT01672866 was reported to be unsuccessful (Gastroenterology, Volume 155, Issue 4, October 2018, Pages 1140-1153).
3) The last updated data in the clinicaltrials.gov NCT00383864 was posted in 2006, etc.
I would like to propose the authors to focus on basic sciences and rewrite as follows;
Kindly include updated description in the sections 1.1-1.4
Section 1.5 may be better to be moved into section 2.
Sections 2.4 and 2.5 could be presented as one independent section.
Sections 3.6 and 3.11 seem to be unnecessary.
Table 1 and its related descriptions can be removed.
In addition, I would like to recommend the authors to have some haptologists read the manuscript. Kindly avoid the inappropriate description such as the interferon therapy for HCV-infected patients.
Author Response
Please see the attachement.

Reviewer 2 Report
The authors tried to review several aspects in liver disease.
I could not understand what the authors want to emphasize for readers.
Hence, it should be condensed and focused on treatment of liver fibrosis, etc..
There are some critical concerns.
Abstract: HPV is not related with cirrhosis.
1.3 chronic hepatitis C: DAA therapy is now standard of care.
3.3 Targeted Molecular Pathways & Drug Delivery
3.5 Monoclonal Antibodies
3.6 Diagnostic Tools
Same topics are duplicate. below
3.8 Targeted Molecular Pathways & Drug Delivery
3.10 Monoclonal Antibodies
3.11 Diagnostic Tools
Round 2
Reviewer 1 Report
The authors responded to the comments. However, I would like to provide minor comments.
The role of the section “Immunology of the Fibrotic Liver” seems to be somehow difficult for readers. I would like to recommend the authors to some minor changes in the title of the section and its sub-sections.
For instance,
2. Immunology of the Fibrotic Liver
→ Inflammation and Progression of the Liver Fibrosis
2.1 Kupffer Cells and Non-Fatty Liver Disease
→ Kupffer Cells and Liver Inflammation in Nonalcoholic Fatty Liver Disease
2.2 T Cell-Mediated Autoimmune Hepatitis
→ T Cell-mediated Inflammation in the Autoimmune Hepatitis
2.3 Naturel Killer and Natural Killer T-cell-related Viral Hepatitis
→ Naturel Killer and Natural Killer T-cell-related Inflammation in Viral Hepatitis
Reviewer 2 Report
The revised manuscript has been improved.
